# Spatial Analysis of Tuberculosis, COVID-19, and Tuberculosis/COVID-19 Coinfection in Recife, PE, Brazil

**DOI:** 10.3390/ijerph22040545

**Published:** 2025-04-02

**Authors:** Alene Bezerra Araújo Silva, Wayner Vieira de Souza, José Constantino Silveira Júnior, Juliana Silva de Santana, Ricardo Arraes de Alencar Ximenes

**Affiliations:** 1Graduate Program in Health Sciences, Faculty of Medical Sciences, University of Pernambuco, Recife CEP 50100-010, PE, Brazil; 2Department of Public Health, Aggeu Magalhães Research Center, Fiocruz, Recife CEP 50.740-465, PE, Brazil; waynervieira@gmail.com; 3Aggeu Magalhães Research Center, Fiocruz, Recife CEP 50.740-465, PE, Brazil; constantinojr10@gmail.com; 4Applied Cellular and Molecular Biology, University of Pernambuco, Recife CEP 50100-010, PE, Brazil; juliana.ssantana3@ufpe.br; 5Department of Tropical Medicine, Federal University of Pernambuco, Recife CEP 50670-901, PE, Brazil; ricardo.ximenes@ufpe.br; 6Department of Medical Clinic, University of Pernambuco, Recife CEP 50100-010, PE, Brazil

**Keywords:** tuberculosis, COVID-19, coinfection, coronavirus, social conditions, risk factors

## Abstract

Tuberculosis (TB) remains a public health problem, which the COVID-19 pandemic may have exacerbated. Scaling TB, COVID-19, and coinfection in area and socioeconomic contexts is an important way to detect more vulnerable groups. Objective: To verify, through the spatial distribution of cases of tuberculosis, COVID-19, and coinfection, the existence of an association between the risk of illness and income. Methods: An analytical ecological study was carried out in Recife, whose unit of analysis was the neighborhood, in the year 2020. The data were collected from the SINAN-TB, NOTIFICA-PE, and IBGE Information Systems. Neighborhoods were grouped into strata according to income through K-means analysis. Incidence rates were calculated. Marshall’s Local Empirical Bayesian Smoothing Method was used. Risk ratios were calculated to estimate the magnitude of association between income strata and incidence rates. Results: A heterogeneous pattern of spatial distribution was verified for the three events addressed, compatible with the inequality of income distribution existing in Recife. For COVID-19, the highest incidence rates occurred in the strata of better-income neighborhoods. There was an association with a gradual increase in the incidence rate as income decreased for tuberculosis. Coinfection did not show a gradual increase in the incidence rate as income decreased, but a lower incidence rate was observed in the stratum of better economic conditions. Conclusions: Studies must be carried out to verify the spatial distribution of COVID-19 and its possible association with socioeconomic factors in subsequent years. There was a positive association between low income and the risk of becoming ill from tuberculosis. The lower incidence rate of coinfection in the stratum of the higher-income population suggests that the pre-existence of TB contributes to illness by COVID-19 in the low-income population.

## 1. Introduction

In 2020, the COVID-19 pandemic began in Brazil, which intensified other public health problems, such as tuberculosis [1].

Until the advent of the COVID-19 pandemic, tuberculosis was the main cause of death by a single infectious agent [1]. Despite the numerous attempts by the World Health Organization to eradicate the bacillus, its incidence rate levels indicate persistence and, sometimes, a recrudescence of the disease over the years, especially in developing countries such as Brazil [1,2]. In Recife, in the northeast of Brazil, the incidence rate of tuberculosis was 84.3 per 100,000 inhabitants in 2020, with 1296 cases of the disease detected [3].

Regarding the risk of illness, the precarious living conditions of individuals in a given population can contribute to its increase due to the association with greater population density, inadequate sanitation, limited access to water and health services, and low education levels, among others. Such context also makes it difficult to implement control measures that, when carried out, can even have their effects diminished [4,5,6,7,8,9].

In contrast to the persistence of tuberculosis, COVID-19 appears quickly and overwhelmingly, incurring a high number of cases and deaths due to complications of infection by the infectious agent SARS-CoV-2 [10,11,12,13,14].

According to the COVID-19 Case Notification System of Pernambuco, NOTIFICA-PE, in Recife, Brazil, during the first year of the pandemic, there were 19,607 cases of the disease, of which 4380 resulted in death [15].

Given the above and the necessity to prioritize the care of COVID-19 cases, routine care was modified. Health teams were directed to individuals affected by SARS-CoV-2, thus constituting obstacles faced in the approach to patients with tuberculosis, which may have led to a decrease in the performance of early diagnosis and timely and adequate treatment [1,16,17].

Still considering the risk of illness, we highlight that the area does not represent only a material geographical surface, in view of the existence of individuals acting on it and developing various activities that modify and compose it. Thus, the area becomes a synthesis of the living conditions of individuals, implying different health risks, as demonstrated by studies related to tuberculosis and COVID-19 [6,8,9,10,11].

In this sense, it is worth highlighting the importance of spatial analysis studies using Geographic Information Systems to better understand the pattern of disease development. During the COVID-19 pandemic, geospatial tools made it possible to assess the influence of a large number of environmental and social factors related to the spread of the virus [18,19].

In view of the association between situations of poverty, area, and illness addressed so far, it is important to highlight that Recife was the capital that, in 2019, presented the highest degree of social inequality, according to the Brazilian Institute of Geography and Statistics (IBGE), with a Gini index of 0.61. Also, according to IBGE data, 6.8% of the population lived below the poverty line in that year, corresponding to approximately 110,000 people [20].

Furthermore, since tuberculosis and COVID-19 are diseases that affect the respiratory system, which can lead to death, the coexistence of tuberculosis and COVID-19 may be a risk factor for the worsening of the condition among those affected by both diseases. Hence, coinfection is a new challenge to be faced by the health system when controlling both diseases [1,16,17].

Thus, this study aimed to verify the association between income and the risk of illness through the analysis of the spatial distribution of cases of tuberculosis, COVID-19, and tuberculosis/COVID-19 coinfection in the neighborhoods of Recife during 2020, the first pandemic year; therefore, the period prior to the introduction of the COVID-19 vaccine.

## 2. Materials and Methods

This is an ecological study of an analytical nature according to spatial aggregates in Recife, whose unit of analysis was neighborhoods. The rationale for the approach adopted in our study is based on the ideas of Milton Santos and assumed, as the work of other Brazilian epidemiologists, that the space captures the interactive processes that permeate the occurrence of health and disease in society and incorporates natural and social determinants [21,22]. The city of Recife, the capital of Pernambuco state, has 94 neighborhoods and is divided into six Political–Administrative Regions (PAR) and eight Health Districts (HD). With approximately 16 million inhabitants and a Gini Index of 0.61, it is a city marked by inequality in income distribution [20,23,24].

The study contains data on reported cases of tuberculosis or COVID-19 in people living in the municipality during 2020, the first year of the pandemic.

Tuberculosis totaled 1296 cases, of which five were excluded due to the lack of information about the neighborhood in the Database of the Information System on Diseases and Notification of Tuberculosis (SINAN-TB) [3].

On the other hand, cases related to COVID-19 totaled 19,607, of which 522 were excluded due to lack of identification of neighborhoods in NOTIFICA-PE, a bank provided by the Center for Strategic Information on Health Surveillance of the Municipality of Recife [15].

Geocoding was necessary to identify the neighborhoods of COVID-19 cases, in view of the absence of neighborhood or area information separately from the address in the database provided.

For cases of coinfection, we used the probabilistic relationship of records with the banks of NOTIFICA-PE and SINAN-TB through the Fine-Grained Record Integration and Linkage (FRIL) program [3,15]. The variables name, date of birth, age, gender, and name of the mother of the patients in both banks were included, and the data were cross-referenced.

Regarding economic and demographic data, a study unit of analysis was obtained by census tract grouped at the neighborhood level, based on the Census carried out in 2010, from IBGE [25]. Subsequently, the neighborhoods were grouped into four income strata, the first two comprising 20% of the population and the last two, 80%, through the K-means analysis, according to the percentage of heads of households with income less than or equal to two minimum wages, comprising income as an estimator of living conditions [26,27]. The choice of variable income was due to its correlation with other socioeconomic variables at the area level (and therefore may be understood as a variable that summarizes the socioeconomic conditions), to the simplicity of working with just one variable, and to the possibility of comparing with other studies of our group [13,26,27].

Thus, the dependent variables were the incidence rate of tuberculosis, COVID-19, and coinfection by neighborhood. The independent variable was the percentage of persons responsible for permanent private households with income up to two minimum wages as a proxy for living conditions.

We calculated the incidence rate for the year 2020 for tuberculosis (per 100,000 inhabitants), COVID-19 (per 1000 inhabitants), and coinfection (per 100,000 inhabitants).

The nominator was the number of new cases in the yearof 2020 and the denominator was the neighborhood population from the Demographic Census of 2010. To calculate the incidence rate for each stratum we added the number of cases and the population of the neighborhoods in each stratum.

In the spatial analysis, Marshall’s Local Empirical Bayesian Smoothing Method was used through the TerraView 4.2.2 software [28], which allowed the smoothing of random fluctuations that may occur, causing distortions in the presentation of information whose units of analysis comprise small areas, such as neighborhoods [26].

Risk ratios were calculated to verify the association between income strata and incidence rates of tuberculosis, COVID-19, and coinfection.

This study used secondary data at the group level; therefore, the cases were not individualized.

## 3. Results

According to the map (Figure 1), the best living conditions associated with income are concentrated in neighborhoods of the central-north region of the city, which presented the lowest percentages of people responsible for permanent households with income up to two minimum wages. To the south, only one neighborhood comprises the highest income stratum.

During 2020, 1296 new cases of tuberculosis were detected in the city of Recife, distributed in the vast majority (1111) among the neighborhoods belonging to strata 3 and 4, with the lowest income, including also the highest incidence rates of the disease, as 89.2 and 93.3 per 100,000 inhabitants, respectively.

The thematic map of the distribution of tuberculosis (Figure 2) forms three major areas of higher incidence rate of the disease in the central and western regions of the city.

Therefore, tuberculosis did not present uniformity in its spatial distribution but a heterogeneous pattern with greater agglomerations in the three large areas mentioned above, in the central and western regions, at points far from each other.

In relation to COVID-19, in 2020, 19,607 cases of the disease were reported, distributed in the vast majority (13,942) among the neighborhoods belonging to strata 3 and 4, with the lowest income. However, the highest incidence rates were observed in strata 1 (15.7 per 1000 inhabitants) and 2 (19.1 per 1000 inhabitants), which have the highest income.

Unlike the tuberculosis distribution map, the COVID-19 map presents a single large area with a higher incidence rate of the disease, located in the central region, and two small areas farther away, in different regions and further west (Figure 3).

Thus, although heterogeneous, the distribution pattern of COVID-19 concentrated the highest incidence rates in a large area, covering mainly neighborhoods located in strata with higher income.

Regarding tuberculosis and COVID-19 coinfection, 133 cases were identified during the study period, mostly distributed (110) in strata 3 and 4. However, the highest incidence rates were observed in strata 2 (12.9 per 100,000 inhabitants) and 3 (9.9 per 100,000 inhabitants).

According to the map (Figure 4), the distribution of coinfection cases is concentrated in two larger areas in the central and western regions of the city, in addition to three smaller areas separated from each other and further west in the city (Figure 4).

Thus, the spatial distribution pattern presented by coinfection is also heterogeneous, with areas of higher incidence rates in different regions, as shown in Figure 4.

The summary table (Table 1) presents the main characteristics found in the strata, with information regarding the number of neighborhoods that formed each stratum, as well as the incidence rates and risk ratios.

As shown in Table 1, a higher incidence rate of tuberculosis was observed in the lowest-income strata. Thus, we notice a gradual increase in the incidence rate among the strata as income decreases.

Stratum 1, with the highest income, presented the lowest incidence rate (44.7 per 100,000 inhabitants). Furthermore, according to the risk ratio, stratum 4, with an incidence rate of 93.3 per 100,000 inhabitants, presented a risk of illness 2.1 times higher than stratum 1.

However, the same association was not demonstrated when observing the spatial distribution of COVID-19, as the incidence rate did not increase as income decreased. The highest incidence rate occurred in the second stratum, with an incidence rate of 19.1 per 1000 inhabitants, followed by the highest income stratum (1), with an incidence rate of 15.7 per 1000 inhabitants, therefore not presenting a gradient between the strata.

Finally, coinfection, as COVID-19, does not present a pattern of spatial distribution compatible with the gradual decrease in income, given that stratum 2 has a higher incidence rate, even when compared to strata 3 (9.9 per 100,000 inhabitants) and 4 (8.4 per 100,000 inhabitants), strata with lower incomes.

However, in stratum 1, with higher income, the lowest incidence (4.9 per 100,000 inhabitants) of coinfection was verified, so the other strata presented higher risks of illness when compared to the stratum with higher income (1).

## 4. Discussion

With the high number of tuberculosis cases, totaling 1296, and an incidence rate of 84.3 per 100,000 inhabitants in the city of Recife, this study pointed to a heterogeneous pattern of distribution and a higher concentration of cases in strata whose neighborhoods comprise those with lower income. Neighborhoods located in areas of the central and western regions of the municipality were the most affected.

However, in other areas of the city, high tuberculosis incidence rates were also found, compatible with unequal income distribution. Studies carried out in other Brazilian cities also show heterogeneous distribution patterns associated with situations of socioeconomic deprivation [29,30,31].

Low income makes up a broader context of less acquisition of goods, less access to adequate housing, population density, and inadequate nutrition, which are considered social determinants of tuberculosis. Thus, income frames are situations considered a risk for the development of tuberculosis [6,7,8,9].

The positive association between tuberculosis and income was verified through the gradual increase in incidence rates concomitant with the decrease in income. Thus, through the calculation of risk ratios, in the last stratum 4, the risk for tuberculosis is 2.1 times higher than in the highest income stratum (1).

Regarding COVID-19, 19,607 cases of the disease were reported, distributed especially in the central region of Recife. Despite the coincident concentration of tuberculosis and COVID-19 cases in this region, the spatial distribution pattern was not the same between the two diseases, considering that, in the case of COVID-19, there was a greater number of neighborhoods with a higher income in the area covered by the highest incidence rates, as shown in Figure 1 and Figure 3. Also, in relation to COVID-19, strata 1 and 2, with the highest income, were the most affected, with incidence rates of 15.7 and 19.1 per 1000 inhabitants, respectively. Although some studies point to the association of the disease with factors of economic vulnerability [10,13,32,33], in this study, the risk of illness did not increase with the decrease in income.

A similar result was found in a study also carried out in the city of Recife with cases of severe acute respiratory syndrome (SARS) in 2020, in which a higher incidence rate of the disease was evidenced in neighborhoods with a lower percentage of precarious settlements during that year [33].

Some factors may have contributed to the absence of an association between COVID-19 and income, of which we can highlight the introduction of SARS-CoV-2, the causative agent of the disease, primarily through populations with greater economic power [13,34]; the study period, the first pandemic year, with rapid contagion and a high number of cases [13,14]; and, finally, without exhausting the possible explanatory causes, greater difficulty in accessing health services by people experiencing poverty, which may have led to a decrease in diagnoses [34,35].

As for the first possible explanatory cause, studies point to the introduction of SARS-CoV-2 by wealthier layers [13,34]. In the state of Ceará, it was observed that the first cases of the disease occurred in the better Municipal Human Development Index (MHDI) neighborhoods of Fortaleza, and low levels of MHDI revealed difficulties in accessing quality health services, causing losses in the diagnosis and treatment of the disease [34].

Other studies carried out in the northeast region of Brazil, particularly in Pernambuco state, also showed higher incidence rates of COVID-19 in different social groups. Thus, although initially the social classes with better living conditions were the most affected, the dissemination and maintenance of the highest incidence rates occurred in social classes with worse living conditions [34,36].

As for the second possible explanatory cause, considering that the introduction of the virus was due to the social classes with better living conditions and that the spread of SARS-CoV-2 occurred very quickly [13,14], it is likely that the number of people affected in this social class was quite high in order to make it difficult to control its spread in this layer, given the lack of effective treatment and corresponding to the period prior to the introduction of the COVID-19 vaccine. In addition, the opposition of the government to social distancing and vaccination aggravated the pandemic scenario, contributing to the increase in the number of cases in the first year of the pandemic [10,13,14,32].

In a study carried out in the city of São Paulo, Brazil, Pinto et al. (2020) verified the formation of several environments and called them “protection bubbles” [37], as a high number of people from the same social group contracted the disease and acquired immunity, relative protection was given to the other people in the group due to the exhaustion of the contagion network. Also, according to this study, social distancing may have been a protective measure in view of the reduction of virus transmission among these different population groups [37]. These protective bubbles are relatively unstable environments that can rupture when new cases of the disease are reintroduced in each group and therefore differ from the expected stability of herd immunity. These protective bubbles can burst, or the networks can be reinitiated; that is, new waves of transmission can occur if social distancing falls too much or there is a reintroduction of the virus in regions of the city where few agents have been infected [37].

Thus, social distancing was responsible for the decrease in the speed of transmission of COVID-19, which initially focused on the layers of better living conditions, migrating to the population groups with the worst living conditions, although flexibility measures have been taken without meeting the minimum criteria and parameters recommended by the who, as pointed out by Ximenes et al. [14].

Finally, in Recife, PE, Brazil, a study carried out in 2020 revealed that, despite the higher mortality rate due to COVID-19 in neighborhoods with a higher percentage of precarious settlements, the registration of SARS cases was higher in neighborhoods with a lower percentage of these settlements, suggesting that the population living in poorer areas may not have had the same access to diagnosis when compared to the richer class [35].

Hence, in addition to corroborating the lack of association revealed by the results presented in this study, this study reinforces that one of the explanatory causes for higher incidence rates in higher-income strata would be the difficulty of accessing health services faced by the poorest sections of society.

In the same vein, a study conducted in the city of Rio de Janeiro found that the neighborhoods in the southern zone of the municipality were the most affected by the disease at the beginning of the epidemic, thus being its potential source of spread, while the neighborhoods in the northern zone, despite having a lower incidence rate, exhibited a higher mortality rate [38].

Therefore, COVID-19 is a disease whose research is still recent, in contrast to tuberculosis research, and should be deepened in order to understand and consolidate the understanding of its spatial distribution and possible association with the respective risk factors.

Thus, the maintenance of research may reveal different scenarios and patterns of the distribution of COVID-19 in subsequent years, in view of the fact that it is an infectious disease, which historically has higher incidence rates in populations with worse living conditions.

In the case of coinfection, as in tuberculosis and COVID-19, the spatial distribution pattern was heterogeneous, with higher incidence rates in areas distributed throughout the central and western regions of the city.

Although it is not possible to demonstrate in this study a gradual increase in cases of coinfection between strata, it was found that there was an increased risk in strata 2, 3, and 4 when compared to the higher-income stratum (1). In addition, strata 3 and 4 correspond to the vast majority of the population of the municipality [26].

Other studies have shown that the pre-existence of chronic diseases leads to risk situations for the development of the severe form of COVID-19 [39,40]. Thus, the lower incidence rate of TB cases in the higher-income stratum may also have contributed to a decrease in the coinfection incidence rate. This pattern is a new challenge for the public health system since the population affected by tuberculosis is at risk for developing severe forms of COVID-19, given the persistence of the disease [40]. Silva et al. (2021) suggest that the interim immunological suppression induced by tuberculosis may increase individuals’ susceptibility to COVID-19 [40].

Recife gathers characteristics that are conducive to the development of infectious diseases, with a marked presence of social inequality. Throughout its history, the city has accumulated incidence rates that exceed the Brazilian average of tuberculosis and was severely affected by the COVID-19 pandemic as one of the main capitals of the northeast. In addition, this study addresses coinfection with tuberculosis and COVID-19, two potentially lethal diseases.

Our study has strengths and limitations. A strength of the study was that it focused on the data of the first pandemic year providing a snapshot of the dynamics of a new person-to-person transmitted infectious disease in a susceptible population, before the start of vaccination. Another point is that, differently from previous studies, it analyzes both conditions (tuberculosis, COVID-19, and tuberculosis–COVID-19 coinfection) producing a comprehensive view of the pattern of the spatial distribution of these two infections that could subsidize the development of more effective public health policies and contribute to the preparation for future health emergencies.

As for the limitations, the “neighborhood” unit of analysis, although widely used by several studies, does not correspond to the ideal unit to work on the distribution of cases of diseases associated with living conditions, in view of the possibility of heterogeneity of these conditions within the same neighborhood. In the literature review on spatial analysis studies addressing COVID-19, it was found that, in the vast majority of cases, statistical techniques and spatial analysis units differed from those used in this study, which hinders the use of these references for comparative purposes [41,42,43,44,45]. Another limitation was that, as the demographic census planned for 2020 was not carried out we had to use the data of a decade-old census (2010).

## 5. Conclusions

In our study, we found no association between low income and higher incidence rates of COVID-19 in the capital of a Northeastern Brazilian State. These findings register a particular moment of the dynamic of the epidemic, after the introduction of the virus in higher income groups and the beginning of its dissemination to poorer areas, but it is representative of what happened in several other Brazilian cities. Studies should be carried out in order to verify its spatial distribution and possible association with living conditions in subsequent years, corresponding to other phases of the epidemic and to the post-epidemic period.

Low income is associated with a higher risk of tuberculosis which highlights the relevance of broader interventions to improving the life conditions of the population and also as a tool to control tuberculosis and several other diseases that are related to social deprivation.

With regard to coinfection, the pre-existence of tuberculosis, particularly in the low-income population, shapes the behavior of tuberculosis and COVID-19 coinfection with higher incidence rates in this population group.

## Figures and Tables

**Figure 1 ijerph-22-00545-f001:**
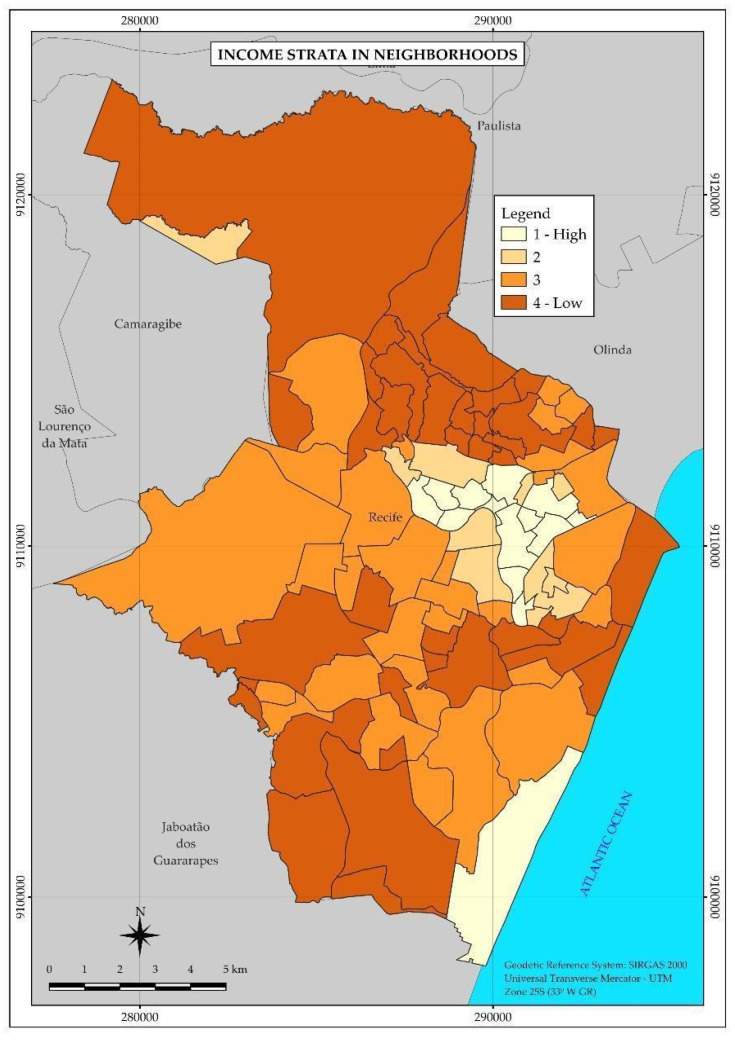
Thematic map of the neighborhoods and income strata of the Municipality of Recife in 2010. Recife, PE, Brazil.

**Figure 2 ijerph-22-00545-f002:**
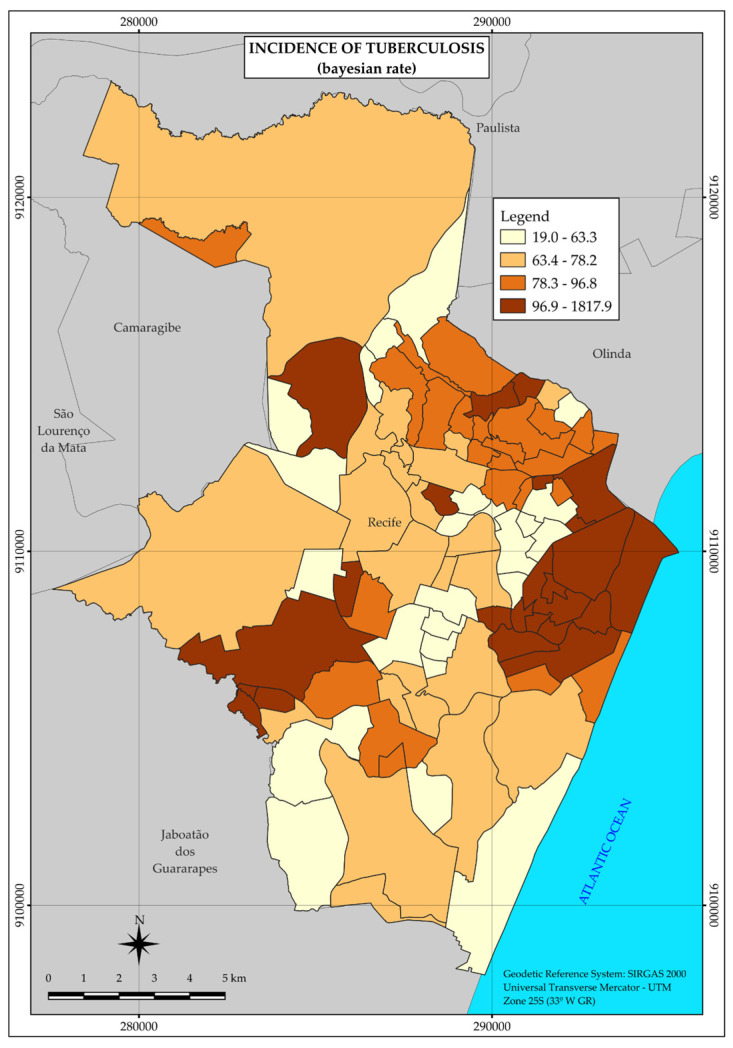
Thematic map of the incidence rate of tuberculosis in the neighborhoods of the Municipality of Recife in 2020. Recife, PE, Brazil.

**Figure 3 ijerph-22-00545-f003:**
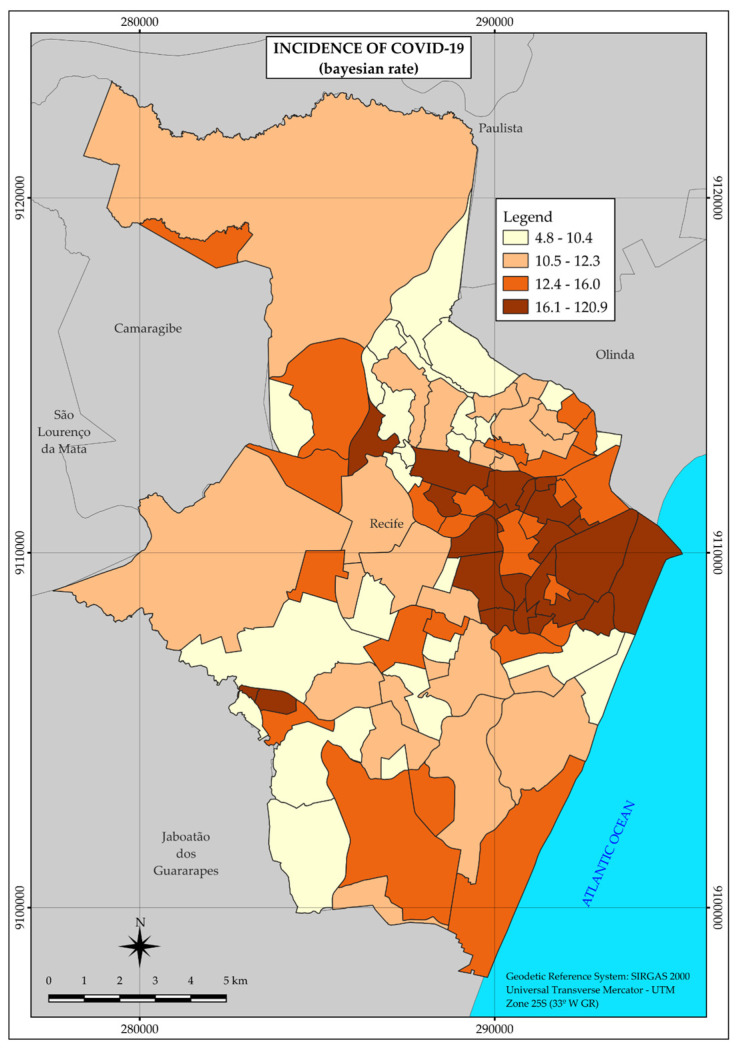
Thematic map of the incidence rate of COVID-19 in the neighborhoods of the Municipality of Recife in 2020. Recife, PE, Brazil.

**Figure 4 ijerph-22-00545-f004:**
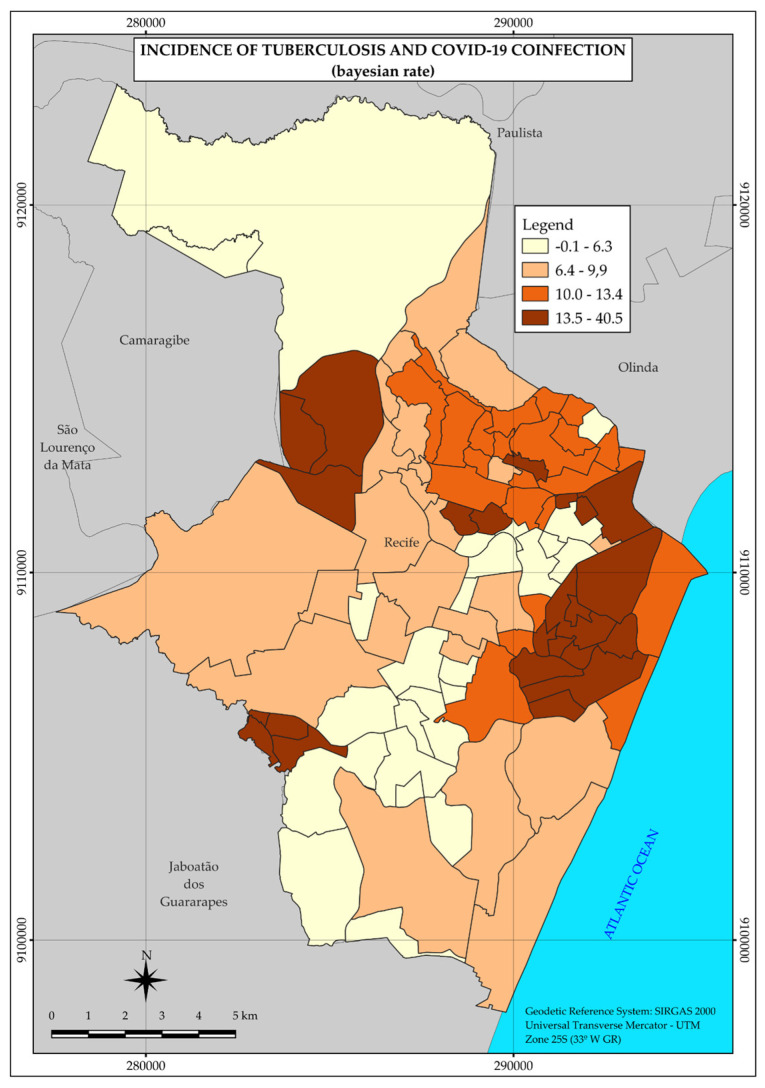
Thematic map of the incidence rate of tuberculosis and COVID-19 coinfection in the neighborhoods of the Municipality of Recife in 2020. Recife, PE, Brazil.

**Table 1 ijerph-22-00545-t001:** Strata of the neighborhoods of Recife with the respective numbers of cases, incidence rate (Inc Rate), and risk ratio (RR) for the diseases of tuberculosis, COVID-19, and coinfection of the diseases in 2020. Recife, PE, Brazil.

Stratum	Neighbor-Hood	Tuberculosis			COVID-19			Coinfection		
		Cases	Inc Rate (Per 100,000 Inhabitants)	RR	Cases	Inc Rate (Per 1000 Inhabitants)	RR	Cases	Inc Rate (Per 100,000 Inhabitants)	RR
1	14	92	44.7	1.0	3231	15.7	1.00	10	4.9	1.00
2	10	88	87.7	2.0	1912	19.1	1.2	13	12.9	2.6
3	30	476	89.2	2	6758	12.7	0.8	53	9.9	2
4	40	635	93.3	2.1	7184	10.6	0.7	57	8.4	1.7

## Data Availability

The datasets presented in this article are not readily available for ethical reasons. Requests for access to datasets should be directed to the City Hall of Recife.

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
