# Peer review of "Spatial Analysis of Tuberculosis, COVID-19, and Tuberculosis/COVID-19 Coinfection in Recife, PE, Brazil"

_ijerph, 2025, doi:10.3390/ijerph22040545_

Round 1

Reviewer 1 Report

Comments and Suggestions for Authors

First of all, congratulations to the authors for their work. I think the subject matter is very interesting. Anyway, I notice 2 weaknesses from the first moment: that it is an issue already worked on a lot and that the data is already a certain age, 5 years have passed. Beyond this, it is an interesting topic, but not an original one. Map 2, 3 and 4 have the wrong legend, you cannot end one interval start the next with the same value. Although they use a territorial delimitation in quite detail, it is a major mistake, since pandemics do not respect administrative limits. Therefore, an entire district is marked in the same color when it would be more appropriate to know the territorial behavior than by streets (for example). They are also very descriptive, they do not explain why one neighborhood has one value and not another, they limit themselves to saying that they are more or less present. A much more thorough analysis is needed before discussion. They wolud have to combine economic data with infrastructures, the presence of hospitals, pharmacies, etc. Many more socioeconomics indicators. In my view it is not enaough. For the paper to be important, they must carry out at least 2 activities: an analysis with the data that they have much more in depth and, secondly, add other socioeconomic variables. 

Author Response

Comments and suggestions for authors (1):  

First of all, congratulations to the authors for their work. I think the subject matter is very interesting. Anyway, I notice 2 weaknesses from the first moment: that it is an issue already worked on a lot and that the data is already a certain age, 5 years have passed.

 Response from the authors (1):

The group agrees with the reviewer regarding the extensive spatial analysis related to tuberculosis, an ancient disease in human history, but one that continues to be a concerning cause of morbidity and mortality in developing countries, such as Brazil. Given that Recife, where the study was conducted, is one of the municipalities with the highest tuberculosis morbidity and mortality rates in the country, the intensification of studies of this nature is justified, especially considering the potential worsening of the situation due to the COVID-19 pandemic.

Regarding spatial analysis studies involving COVID-19, during the literature review, it was noted that there is a scarcity of studies adopting this approach, particularly those that simultaneously analyse both conditions.

It is crucial to address both conditions, as they are infectious diseases and, historically, infectious diseases have shown a strong association with poverty situations. Thus, considering that space is a strong indicator of socioeconomic conditions, it is important to understand the spatial distribution of tuberculosis, COVID-19, and tuberculosis-COVID-19 co-infection for the planning of more effective policies in the municipality—policies that could serve as a model for other localities in similar situations.

It is also worth noting that Recife, in 2019, the period immediately preceding the start of the pandemic, was the capital with the highest degree of social inequality, according to IBGE (Brazilian Institute of Geography and Statistics), with a Gini index of 0.61. Additionally, 6.8% of the population, approximately 110,000 people, lived below the poverty line that year. This scenario suggests an unequal distribution of infectious disease cases, which significantly hinders the control of both diseases, tuberculosis and COVID-19, in the municipality.

Another factor to consider is that both tuberculosis and COVID-19 affect organs of the respiratory system, and the concurrent presence of these diseases in an individual can result in more severe conditions. This reinforces the need for ecologic studies using spatial analysis to investigate the distribution of the incidence of both infections.

Comments and Suggestions for Authors (2):

First of all, congratulations to the authors for their work. I think the subject matter is very interesting. Anyway, I notice 2 weaknesses from the first moment: that it is an issue already worked on a lot and that the data is already a certain age, 5 years have passed.

Response from the authors 2:

The group understands the importance of addressing data from the pandemic period, considering the high number of COVID-19 cases at that time and its relevance to the development of the research. Our choice was to study the year of 2020, as it was the first pandemic year, therefore, the period prior to the introduction of the COVID-19 vaccine. It allows the understanding of the dynamics of a new person-to-person transmitted infectious disease in a population. The large number of COVID-19 cases reflects a situation that may not be explored in the same way in subsequent periods, after the start of vaccination and decrease of the number of susceptible individuals. This period brought valuable contributions to science and, as has happened at other moments in history, it is possible that humanity will face similar situations in the future.

It is necessary to have records that help in understanding the behavior of diseases during major crises, such as the one caused by COVID-19. These records are essential for the development of more effective public health policies and for preparation for future health emergencies.

Comments and Suggestions for Authors (3):

Map 2, 3 and 4 have the wrong legend, you cannot end one interval start the next with the same value.

Response from the authors (3):

The intervals are automatically defined by the Terraview software used in the study, where the upper limit is not included in the interval. The team agrees with the reviewer that the data, as presented, may cause confusion and misinterpretation in conveying the information to readers and has chosen to modify the map captions, as this is allowed by the software during the map-building process. Thus, new maps were generated with the suggested adjustment.

Comments and Suggestions for Authors (4):

Although they use territorial delimitation in quite detail, it is a major mistake, since pandemics do not respect administrative limits. Therefore, an entire district is marked in the same color when it would be more appropriate to know the territorial behavior than by streets (for example).

Response from the authors (4):

The group of authors used the neighborhood as the unit of analysis to verify the incidences and distribution of diseases in the municipality of Recife. Recife has 94 neighborhoods, which were grouped into income strata and graphically represented. Similarly, for better visualization of the disease distribution, maps of the health conditions were presented according to their incidences. Just as with income, the health conditions were also divided into neighborhood groups (quartiles) based on the observed incidences.

Therefore, the colors in the graphs vary according to the neighborhood groups in relation to income and the incidences of tuberculosis, COVID-19, and co-infection. This way, it is possible to graphically observe whether there was an overlap between the neighborhood groups with the highest number of disease cases and those with the lowest income, or whether no such relationship existed. This is why the graphs were designed with colors based on strata rather than individual neighborhoods or streets.

Comments and Suggestions for Authors (5):

They are also very descriptive, they do not explain why one neighborhood has one value and not another, they limit themselves to saying that they are more or less present

Response from the authors (5):

It is important to highlight how the income dimension is intrinsically related to the composition and spatial characterization in the municipality of Recife. According to Jan Bitoun, when comparing the variation of the HDI (Human Development Index) between regions in the municipality, it was found that, among the dimensions of longevity, education, and income, it is in the latter that Recife exhibits the greatest variation  within municipalities. He also pointed out that, in the 1990s, the range of differences in income values between the highest and lowest regions increased, with a 6.1% rise in inequalities between the regions.

As will be explained in more detail in the following responses, the concept of space encompasses aspects that reflect the living conditions of a population, with income being one of the main factors. Low income, in turn, contributes to precarious living conditions, which may be associated with a higher risk of illness.

All these elements can be considered components that make up space, and the relationship between space, income, living conditions, and the risk of illness was addressed throughout the article. However, the discussion was not conducted individually for each neighborhood, which constituted the unit of analysis of the study. Therefore, below are some excerpts from the approach to this theme in the article:

“Regarding the risk of illness, the precarious living conditions of individuals in a given population can contribute to its increase due to the association with greater population density, inadequate sanitation, limited access to water and health services, low education levels, among others. Such context also makes it difficult to implement control measures that, when carried out, can even have their effects diminished [4-9].” (lines 49 - 53)

“Low income makes up a broader context of less acquisition of goods, less access to adequate housing, population density, and inadequate nutrition, which are considered social determinants of tuberculosis. Thus, income frames are situations considered a risk for the development of tuberculosis [6-9].” (lines 234 - 237)

Comments and Suggestions for Authors (6):

A much more thorough analysis is needed before discussion. They would have to combine economic data with infrastructures, the presence of hospitals, pharmacies, etc. Many more socioeconomics indicators. In my view it is not enough. For the paper to be important, they must carry out at least 2 activities: an analysis with the data that they have much more in depth and secondly, add other socioeconomic variables.

 Response from the authors (6):

The approach adopted in our study is influenced by the concept of territory and space proposed by Milton Santos [1,2], which was adopted by several Brazilian epidemiologists to understand the spatial behavior of the health and disease process. According to this author the concept of space incorporates natural and social determinants; the space captures the interactive processes that permeate the occurrence of health and disease in society [1,2]. As pointed out by Silva et al. (1997) [3] this approach allowed the epidemiology to move from the analysis focused on the disease to the analysis of the conditions of occurrence of the diseases. Therefore, assuming this rationale, we would prefer not to fragment the space into some of its various components (like infrastructures, the presence of hospitals, pharmacies, etc) but to consider it in its entirety, assuming that it synthetizes the conditions and situations determinants, at the area level, of the risk of the disease. We are aware of the relevance of the analysis suggested by the reviewer, but it corresponds to another approach which is out of the scope of this paper.

 1- Faria, R. M., & Bortolozzi, A. (2009). Espaço, território e saúde: contribuições de Milton Santos para o tema da geografia da saúde no Brasil. Raega-O Espaço Geográfico em Análise, 17, 31-41.

2- Saquet, M. A., & da Silva, S. S. (2011). MILTON SANTOS: concepções de geografia, espaço e território MILTON SANTOS: Geography conceptions, space and territory. Geo UERJ. ano, 10, 24-42.

3- SILVA, Luis Jacinto. The concept of space in infectious disease epidemiology. Cadernos de Saúde Pública, Rio de Janeiro, v. 13, n. 4, p. 585-593, 1997.

Comments and Suggestions for Authors (7):

A much more thorough analysis is needed before discussion. They would have to combine economic data with infrastructures, the presence of hospitals, pharmacies, etc. Many more socioeconomics indicators. In my view it is not enaough. For the paper to be important, they must carry out at least 2 activities:  an analysis with the data that they have much more in depth and secondly, add other socioeconomic variables.

Response from the authors (7):

Our group has been accumulating experience with ecological studies of infectious diseases in urban areas, having worked with tuberculosis [1,2,3], leprosy [4], filariasis [5],  hepatitis [6] and Zika [7, 8], Dengue, Chikungunya and Zika [9] and COVID [10]. To study the association between the socioeconomic conditions at the area level and the frequency of the diseases we used different approaches, either i) independent socioeconomic variables [2,3, 4, 6,7,8,9,10], ii) synthesis of several variables in one utilizing different methodological strategies, i.e., factor analysis [1],  construction of a deprivation index [4] or a social and environmental indicator [6]. In the paper of Lapa et al. (2001) we compared two different strategies, the use of a deprivation index (composed of income, education, water supply, inhabitants per bedroom, housing in substandard clusters and population density) and income  to discriminate strata of high, intermediate and low risk for leprosy and we found an agreement between the two classifications for 96.7% of the total census tracts studied. Taking into account this finding and assuming that the variable income at the area level is correlated with other socioeconomic variables and therefore may be understood as a  variable that summarizes the socioeconomic conditions, and for the simplicity of working with just one variable our option, in the more recent studies [7,8,9,10]  has been to work with income at the area level, and we would prefer to keep income in this paper. In addition, it makes it possible the comparison of this paper with previous studies carried out in the same urban area [7,8,9,10]

 1- Souza WV, Ximenes R, Albuquerque MFM, Lapa TM, Portugal JL, Lima MLC et al. The use of socioeconomic factors in mapping tuberculosis risk areas in a city of northeastern Brazil. Rev Panam Salud Publica 2000;8:403-10.

2- Souza WV, Albuquerque Mde F, Barcellos CC, Ximenes RA, Carvalho MS. Tuberculose no Brasil: construção de um sistema de vigilância de base territorial [Tuberculosis in Brazil: construction of a territorially based surveillance system]. Rev Saude Publica. 2005 Feb;39(1):82-9. Portuguese. doi: 10.1590/s0034-89102005000100011. Epub 2005 Jan 11. PMID: 15654464.

3- Souza WV, Carvalho MS, Albuquerque Mde F, Barcellos CC, Ximenes RA. Tuberculosis in intra-urban settings: a Bayesian approach. Trop Med Int Health. 2007 Mar;12(3):323-30. doi: 10.1111/j.1365-3156.2006.01797.x. PMID: 17286622.

4- Lapa T, Ximenes R, Silva NN, Souza W, Albuquerque M, Campozana G. Vigilância da hanseníase em Olinda, Brasil, utilizando técnicas de análise espacial [Leprosy surveillance in Olinda, Brazil, using spatial analysis techniques]. Cad Saude Publica. 2001 Sep-Oct;17(5):1153-62. Portuguese. doi: 10.1590/s0102-311x2001000500016. PMID: 11679890.

5- Braga C, Ximenes RA de A, Albuquerque M de FPM de, Souza W Vde, Miranda J, Brayner F et al. Evaluation of a social and environmental indicator used in the identification of lymphatic filariasis transmission in urban centers. Cad Saúde Pública 2001;17:1211-8.

6 - de Alencar Ximenes RA, Martelli CM, Merchán-Hamann E, Montarroyos UR, Braga MC, de Lima ML, Cardoso MR, Turchi MD, Costa MA, de Alencar LC, Moreira RC, Figueiredo GM, Pereira LM; Hepatitis Study Group. Multilevel analysis of hepatitis A infection in children and adolescents: a household survey in the Northeast and Central-west regions of Brazil. Int J Epidemiol. 2008 Aug;37(4):852-61. doi: 10.1093/ije/dyn114. PMID: 18653514; PMCID: PMC24833

7- Souza WV, Albuquerque MFPM, Vazquez E, Bezerra LCA, Mendes ADCG, Lyra TM, Araujo TVB, Oliveira ALS, Braga MC, Ximenes RAA, Miranda-Filho DB, Cabral Silva APS, Rodrigues L, Martelli CMT. Microcephaly epidemic related to the Zika virus and living conditions in Recife, Northeast Brazil. BMC Public Health. 2018 Jan 12;18(1):130. doi: 10.1186/s12889-018-5039-z. PMID: 29329574; PMCID: PMC5767029.

8- Lobkowicz L, Power GM, De Souza WV, Montarroyos UR, Martelli CMT, de Araùjo TVB, Bezerra LCA, Dhalia R, Marques ETA, Miranda-Filho DB, Brickley EB, Ximenes RAA. Neighbourhood-level income and Zika virus infection during pregnancy in Recife, Pernambuco, Brazil: an ecological perspective, 2015-2017. BMJ Glob Health. 2021 Dec;6(12):e006811. doi: 10.1136/bmjgh-2021-006811. PMID: 34857522; PMCID: PMC8640636.

9- Braga C, Martelli CMT, Souza WV, Luna CF, Albuquerque MFPM, Mariz CA, Morais CNL, Brito CAA, Melo CFCA, Lins RD, Drexler JF, Jaenisch T, Marques ETA, Viana IFT. Seroprevalence of Dengue, Chikungunya and Zika at the epicenter of the congenital microcephaly epidemic in Northeast Brazil: A population-based survey. PLoS Negl Trop Dis. 2023 Jul 3;17(7):e0011270. doi: 10.1371/journal.pntd.0011270. PMID: 37399197; PMCID: PMC10348596.

10- Souza WV, Martelli CMT, Silva APSC, Maia LTS, Braga MC, Bezerra LCA, Dimech GS, Montarroyos UR, Araújo TVB, Barros Miranda-Filho D, Ximenes RAA, Albuquerque MFPM. The first hundred days of COVID-19 in Pernambuco State, Brazil: epidemiology in historical context. Cad Saude Publica. 2020 Dec 18;36(11):e00228220. Portuguese, English. doi: 10.1590/0102-311X00228220. PMID: 33331595.

Reviewer 2 Report

Comments and Suggestions for Authors

This paper presents the geo-spatial overview of the incidence of TB, COVID and TB-COVID co-infection in a Brazilian city in 2020. The study assess the distribution of disease by socio-economic strata in a town of large economic disparity, showing the associations of TB with low SEC areas of town, while COVID was typically associated with higher SEC areas in the early part of the pandemic. While the analysis is simple in approach, this is an interesting and topical research study, that is worth publishing. However, I do think  revisions are needed before this.

Major comments:

  • Incidence and incidence rates are used synonymously/inconsistently throughout the manuscript. The denominator and/or time frame are not always provided, making the data hard to interpret. eg p2, line45,46 and p8, line 192, 193
  • Methods: do the TB and covid databases used collect data from both private and public health sector?
  • Discussion: Reference 35 does not address TB specifically that I could see on a quick look over. I am not able to read ref 36, as its not in English.  While co-infected patients have a higher mortality, I am not clear that TB increases your risk of Covid.
  • P9 paragraph 6 "It is noteworthy..." I am not clear how this paragraph contributes to the discussion of your study results. 
  • Limitations: this section is very thin. Other limitations surely include: the use of a decade old census for your denominators, changes in reporting patterns for TB disease in 2020
  • Conclusion: it rational for your last comment regarding the role of TB in co-infection is not clear to me: surely this is finding is simply a reflection of the relationship between TB and lower socio-economic strata

Minor comments:

  • P2 lines45, 46: you refer to infection rate and then disease: where is the infection rate from? Or did you mean incidence?
  • P9 line 255, 256: I am not sure what you mean by "mistaken confrontation of the governments?"
  • Careful editing is required: eg P9, line 269: WHO should be in capitals
  • Reference formating is inconsistent through the manuscript.
Comments on the Quality of English Language

The English can be simplified to make this easier to read. 

Author Response

Comments and Suggestions for Authors (1):  

Incidence and incidence rates are used synonymously/inconsistently throughout the manuscript. The denominator and/or time frame are not always provided, making the data hard to interpret. eg p2, line 45, 46 and p8, line 192, 193.

Response from the authors (1):

We calculated the incidence rate for the year of 2020 for tuberculosis (per 100,000 inhabitants), COVID-19 (per 1,000 inhabitants) and coinfection (per 100,000 inhabitants). 

The nominator was the number of new cases in the year of 2020 and the denominator was neighborhood population from the Demographic Census of 2010. To calculate the incidence rate for each stratum we added the number of cases and the population of the  neighborhoods in each stratum. 

A limitation of the study was that, as the demographic census planned for 2020 was not carried out by the Brazilian Government, we had to use the data of demographic census of 2010.

We corrected the inconsistency in the use of incidence and incidence rates throughout the manuscript. 

The information on the calculation of the incidence rates was added to the Methods - (page 3, paragraphs 7 and 8,  lines 128 - 133)

Comments and Suggestions for Authors (2):

Methods: do the TB and covid databases used collect data from both private and public health sector? 

Response from the authors (2):

The databases used encompass all healthcare establishments, both public and private. 

Tuberculosis is a disease of compulsory notification in Brazil. Notification to the Surveillance System for Infectious Diseases (SINAN/MS) is a prerequisite for initiation of TB treatment which is only carried out by the public sector.

Comments and Suggestions for Authors (3):

Discussion: Reference 35 does not address TB specifically that I could see on a quick look over. 

Response from the authors (3):

Reference 35 was restricted to the first part of the paragraph, where tuberculosis was not mentioned, but it was only stated that pre-existing diseases suggest a higher likelihood of developing the severe form of COVID-19. This reference was removed from the second part of the paragraph, which specifically addresses tuberculosis.

Comments and Suggestions for Authors (4):

I am not able to read ref 36, as its not in English.  While coinfected patients have a higher mortality, I am not clear that TB increases your risk of Covid.

Response from the authors (4):

Silva et al. (2021) suggest that the interim immunological suppression induced by tuberculosis may increase individuals' susceptibility to COVID-19.

This information was included in the article's text - page 10, 

Comments and Suggestions for Authors (4):

I am not able to read ref 36, as its not in English.  While coinfected patients have a higher mortality, I am not clear that TB increases your risk of Covid.

Response from the authors (4):

Silva et al. (2021) suggest that the interim immunological suppression induced by tuberculosis may increase individuals' susceptibility to COVID-19.

This information was included in the article's text on page x paragraph 6, lines 330-332.

Comments and Suggestions for Authors (5):

P9 paragraph 6 "It is noteworthy..." I am not clear how this paragraph contributes to the discussion of your study results.

Response from the authors (5):

We think that information given in P9 paragraph 6 is important to differentiate  exhaustion of the contagion network from herd immunity theory. We agree that the text was confusing. To make this point clearer we changed the text and replaced the paragraph 6 to close to the information on contagion network  in line 288 - page 9, paragraph 6, lines 289-294.

Comments and Suggestions for Authors (6):

Limitations: this section is very thin. Other limitations surely include: the use of a decade old census for your denominators, changes in reporting patterns for TB disease in 2020 

Response from the authors (6):

We added the limitation of the use of a decade old census for your denominators. The demographic census planned for 2020 was not carried out, so we had to use the data of the demographic census of 2010. However, the demographic dynamic did not produce important  changes in the neighborhood population - page 10, paragraph 8, lines 350-352.

Comments and Suggestions for Authors (7):

Conclusion: it rational for your last comment regarding the role of TB in co-infection is not clear to me: surely this is finding is simply a reflection of the relationship between TB and lower socio-economic strata

Response from the authors (7):

We changed the text - page 11, paragraph 1, lines 359-361.

Minor comments: 

Comments and Suggestions for Authors (8):

P2 lines45, 46: you refer to infection rate and then disease: where is the infection rate from? Or did you mean incidence? 

Response from the authors (8):

We changed the text - page 2, paragraph 1, line 47

Comments and Suggestions for Authors (9):

P9 line 255, 256: I am not sure what you mean by "mistaken confrontation of the governments?" 

Response from the authors (9):

We changed the text to make this point clearer - page 9, paragraph 5, lines 279-281.

Comments and Suggestions for Authors (10):

Careful editing is required: eg P9, line 269: WHO should be in capitals 

Comments and Suggestions for Authors (11):

We corrected the text - page 9, paragraph 7, lines 299.

Round 2

Reviewer 1 Report

Comments and Suggestions for Authors

First of all, it is appreciated that they have taken into account all the proposed recommendations with the sole objective of improving, as far as possible, their work.

As in the previous review, I congratulate them on their work. In addition, although I continue to detect weaknesses, which I will comment, the work has improved a lot.

Now, I think they should focus on improving 2 aspects.

  1. Increase references to studies on territorial analysis of the pandemic. They can use jobs from other areas such as Europe, China, etc.
  2. Conclusions cannot be limited to 6 lines of text. For example, they should relate to Brazil, to a state, to other territories, etc., which allows them to provide security for their work.

Carballada, A.M.; Balsa-Barreiro, J. Geospatial Analysis and Mapping Strategies for Fine-Grained and Detailed COVID-19 Data with GIS. ISPRS Int. J. Geo-Inf. 2021, 10, 602. https://doi.org/10.3390/ijgi10090602

Author Response

Comments 1:

Now, I think they should focus on improving 2 aspects.

  1. Increase references to studies on territorial analysis of the pandemic. They can use jobs from other areas such as Europe, China, etc.

Response 1:

The group agrees with the considerations and conducted a new literature search on spatial studies addressing COVID-19 and the following references were added in page 2, paragraph 8, lines 70-74 and page 10, paragraph 2, lines 314-318:

  1. Carballada, A.M.; Balsa-Barreiro, J. Geospatial Analysis and Mapping Strategies for Fine-Grained and Detailed COVID-19 Data with GIS. ISPRS Int. J. Geo-Inf. 2021, 10, 602. https://doi.org/10.3390/ijgi10090602
  2. Fatima M, O'Keefe KJ, Wei W, Arshad S, Gruebner O. Geospatial Analysis of COVID-19: A Scoping Review. Int J Environ Res Public Health. 2021 Feb 27;18(5):2336. doi: 10.3390/ijerph18052336. PMID: 33673545; PMCID: PMC7956835.
  3. Cavalcante, J.R.; Abreu, A.J.L. COVID-19 in the city of Rio de Janeiro: Spatial analysis of first confirmed cases and deaths. Epidemiol. Serv. Saude 2020, 29, 9

During the literature search on spatial analysis studies addressing COVID-19 we identified other studies (in addition to those listed above), however, in the vast majority of cases, the statistical techniques and spatial units of analysis used differed from those utilized in our study, which limits the use of these references for comparative purposes:.

  1. Andrade, L.A.; Gomes, D.S.; Góes, M.A.; Souza, M.S.; Teixeira, D.C.; Ribeiro, C.J.; Alves, J.A.; Araújo, K.C.; Santos, A.D. Surveillance of the first cases of COVID-19 in Sergipe using a prospective spatiotemporal analysis: The spatial dispersion and its public health implications. Rev. Soc. Bras. Med. Trop. 2020, 53, 0037–8682.

43 Huang, R.; Liu, M.; Ding, Y. Spatial-temporal distribution of COVID-19 in China and its prediction: A data-driven modeling analysis. J. Infect. Dev. Ctries 2020, 14, 246–253.

  1. Yang, W.; Deng, M.; Li, C.; Huang, J. Spatio-Temporal Patterns of the 2019-nCoV Epidemic at the County Level in Hubei Province, China. Int. J. Environ. Res. Public Health 2020, 17, 2563.
  2. Macharia, P.M.; Joseph, N.K.; Okiro, E.A. A vulnerability index for COVID-19: Spatial analysis at the subnational level in Kenya. BMJGlob. Health 2020, 5, e003014.
  3. Mollalo, A.; Vahedi, B.; Rivera, K.M. GIS-based spatial modeling of COVID-19 incidence rate in the continental United States. Sci. Total Environ. 2020, 728, 22.

Comments 2:

Now, I think they should focus on improving 2 aspects.

2. Conclusions cannot be limited to 6 lines of text. For example, they should relate to Brazil, to a state, to other territories, etc., which allows them to provide security for their work.

Response 2:

The conclusion was changed, as suggested.

Reviewer 2 Report

Comments and Suggestions for Authors

a

Author Response

We thank you for the contributions.